# Identifying Different Mutation Sites Leading to Resistance to the Direct-Acting Antiviral (DAA) Sofosbuvir in Hepatitis C Virus Patients from Egypt

**DOI:** 10.3390/microorganisms10040679

**Published:** 2022-03-22

**Authors:** Aly Atef Shoun, Rania Abozahra, Kholoud Baraka, Mai Mehrez, Sarah M. Abdelhamid

**Affiliations:** 1Microbiology and Immunology Department, Faculty of Pharmacy, Sinai University, El Arish 45518, Egypt; 2Microbiology and Immunology Department, Faculty of Pharmacy, Damanhour University, Damanhour 22511, Egypt; rania.ragab@pharm.dmu.edu.eg (R.A.); kholoud.baraka@pharm.dmu.edu.eg (K.B.); sara.magdy@pharm.dmu.edu.eg (S.M.A.); 3National Hepatology and Tropical Medicine Research Institute (NHTMRI), Cairo 11511, Egypt; maymehrezmd@yahoo.com

**Keywords:** HCV, RNA-dependent RNA polymerase, RNA sequencing, resistance, molecular dynamics

## Abstract

The hepatitis C virus (HCV) is a major global health challenge and a leading cause of morbidity and mortality. Many direct-acting antivirals (DAAs) target essential macromolecules involved in the virus’ life cycle. Although such DAAs achieve great success in reducing the viral load in genotype 1 infections, other genotypes demonstrate different levels of response. This study focused on mutation sites associated with patients with genotype 4a infections that failed to respond to treatment with sofosbuvir. The genotyping of HCV samples from patients with virological failure, and responder patients, was conducted using Geno2Pheno webserver-based full NS5B sequences. We constructed 3D structural models for all the samples and used structural analysis to investigate the effect of amino acid substitution on the observed resistance to SOF-based treatment, and the docking of sofosbuvir into the active sites of the 10 models was performed. Finally, 10 molecular dynamic (MD) simulation experiments were conducted to compare the stability of the 3D models of the resistant samples against the stability of the 3D models of the responder samples. The results highlighted the presence of HCV subtype 4a in all ten samples; in addition, an amino acid (aa) substitution in the palm region may hinder HCV polymerase activity. In this study, we provide evidence that a mutation in the NS5B gene that induces resistance to sofosbuvir in patients with the S282T/C/R mutant virus is present in the Egyptian population. Overall, the docking and MD results support our findings and highlight the significant impact of the identified mutations on the resistance of HCV NS5B RNA-dependent RNA polymerase to direct-acting antivirals (DAAs).

## 1. Introduction

HCV is a global health threat, with more than 71 million cases annually, and is the primary cause of death for nearly 399,000 people each year [1]. Chronic HCV infections are associated with the development of liver cirrhosis, hepatocellular carcinoma, liver failure, and death [1,2]. 

HCV is an enveloped single-stranded, positive-sense RNA virus (+ssRNA virus) belonging to the Flaviviridae family and the *Hepacivirus* genus [3]. Its genome is approximately 9.6 kb in length, and it shows genetic diversity, with eight confirmed genotypes, numbered 1 to 8, and 86 subtypes [4,5]. HCV genotyping plays a crucial role in molecular epidemiology and aids in determining appropriate DAAs and treatment durations [6]. Genotype 1 of HCV is the most common type in the United States, Europe, and Japan, accounting for approximately 60% of total global infections [7]. A high incidence of genotype 2 is found in Western Africa [7], whereas HCV GT3 predominates in Southeast Asia. Genotype 4 is broadly disseminated in Egypt, with limited infections elsewhere in the world. Genotypes 5, 6, and 7 are mainly prevalent in South Africa, Southeast Asia, and Central Africa, respectively. Egypt shows the highest prevalence of HCV globally, with nearly 15% of the total population suffering from HCV, and genotype 4a is the most common subtype [8,9]; this is attributable to improper parenteral injections during antischistosomiasis campaigns that continued from the 1920s to 1980s [10].

The HCV genome encodes a polyprotein of nearly 3000 amino acids, divided into 10 viral proteins. Seven of these proteins are non-structural: P7, NS2, NS3, NS4A, NS4B, NS5A, and NS5B. The three structural proteins are Core, envelope E1, and E2 [11]. HCV non-structural 5B (NS5B), an RNA-dependent RNA polymerase, comprises ~591 amino acids and located at the C-terminus of the polyprotein chain; it is responsible for the replication of the positive-strand genomic RNA [12,13,14]. Many inhibitors that target this enzyme are available. The pan-genotypic inhibitor sofosbuvir initiated a new era in the treatment of HCV infections after its approval in 2013 [15]. RNA-dependent RNA polymerase (RdRp) lacks proofreading ability, which may be the main source of mutations in the HCV genome [16]. HCV NS5B has no match in mammalian cells and is thus a validated target for antiviral therapy with few target-related side effects. HCV NS5B is 65 KDa, with characteristic palm, thumb, and finger domains [17]. The palm domain contains the active site, in addition to four allosteric sites: palm I, palm II, thumb I, and thumb II [17].

Several DAAs for treating HCV have been developed in the last eight years as an effective, safe, and well-tolerated combination with very high sustained virological responses (SVRs)—defined as HCV RNA becoming undetectable 12 or 24 weeks after the end of treatment [18]. After the successful treatment of HCV-infected patients with oral DAA, the level of HCV RNA remains undetectable [19]. However, in some patients, the DAA does not completely eradicate the HCV RNA [20]. A DAA’s activity varies among HCV genotypes due to structural differences among the protein targets [20]. The variable response is mainly due to drug-resistance-associated substitution (RAS). RAS is found in quasi-species in some HCV-infected patients. These variants may not be sensitive to DAAs [21]. Most data on HCV drug resistance relate to genotypes 1 and 3; however, genotype 4 infects approximately 18 million people in the Middle East and North Africa and is not sufficiently studied [22,23]. 

The purpose of this study was to identify the mutations responsible for resistance to sofosbuvir-based treatment for HCV genotype 4a, and to characterize their role in resistance to sofosbuvir using homology modeling, docking, and molecular dynamic simulations. 

## 2. Patients and Methods

### 2.1. Blood Sampling

Thirty-four samples were collected from the National Hepatology and Tropical Medicine Research Institute. Informed consent was obtained from each patient before taking the sample. All the available samples and patient data were collected with informed ethical consent and Institutional Review Board (IRP) approval. The Damanhur Local Ethics Committee approved the study (code 1121PM24F) and administered the ethical guidelines from the 1975 Declaration of Helsinki.

Initially, 14 males and 20 females, aged 41–79 years, with viral loads ranging from 307,000 to 18,969,000 IU/L determined using a COBAS^®^ AmpliPrep/Quantitative Test (v2.0), following the manufacturer’s instructions, were included. Other serum samples from the same patients were collected and were centrifuged at 6000 RPM within 2 h of sample collection; 0.5–1 mL of serum was transferred using a sterile pipette into a clean, sterile, previously labeled 1.5 mL Eppendorf tube.

### 2.2. Viral RNA Extraction

The RNA extraction used an Invitrogen Viral RNA kit (Invitrogen, Thermofisher Scientific, Waltham, MA, USA), following the manufacturer’s instructions. The RNA was subsequently reverse transcribed. For the cDNA preparation, a High-Capacity cDNA Reverse Transcription Kit (Thermofisher Scientific, Waltham, MA, USA) was used, following the manufacturer’s instructions, except for adding primers. We used 2 μL of a random primer, 2 μL of forward primer region 1, and 2 μL of reverse primer region 2.

### 2.3. Primer Sequences and PCR Program

The NS5B region size is approximately 1773 bp. Therefore, it was impossible to amplify and sequence the entire region as one fragment. Instead, we implemented an overlapping amplicon strategy to cover the entire sequence of this region (Table 1). 

### 2.4. PCR Amplification of the NS5B Region

A specific PCR product was designed to amplify the entire NS5B region from the synthesized cDNA. The PCRs included 25 μL of a DreamtaqHot Start PCR master mix kit (Thermo Scientific, Waltham, MA, USA), 2 μL of a random primer, 2 μL of the forward primer, 2 μL of the reverse primer, and 4 μL of each cDNA sample. The reaction was made up to a final volume of 50 μL with water. The reactions were loaded into a thermal cycler (Multigene, Labnet, Edison, NJ, USA) under the following conditions: for the Maxima Hot Start PCR master mix kit, an initial enzyme activation for 5 min at 95 °C was followed by 38 cycles of 45 s at 95 °C for denaturation, 45 s at a selected annealing temperature, and 1–2 min at 72 °C for extension. The final extension was performed for 10 min at 72 °C. The PCR amplification product was electrophoresed on an agarose gel before PCR purification with a 1.8% agarose gel in TAE buffer. A 100 bp DNA ladder was used (Appendix A). 

### 2.5. Multiple Sequence Alignment

A total of 2202 HCV whole-genome sequences were aligned using the MUSCEL MSA tool, and the resulting file was used in the design of primers. We focused only on genotype 4a, the genotype that displays the highest incidence in Egypt.

### 2.6. Sanger Sequencing

NS5B fragments were amplified using a GeneJet PCR purification kit (K0691, Thermo Scientific, Waltham, MA, USA). The purified amplicons were sequenced on both strands via bi-directional Sanger sequencing using a 3730 xl DNA analyzer.

### 2.7. Data Analysis

Data analysis was performed using the Geno2Pheno webserver.

### 2.8. Phylogenetic Analysis

The evolutionary distances were calculated using the Kimura 2-parameter method with a discrete gamma distribution using the MEGA-X software, and a phylogenetic tree was constructed using the neighbor-joining method. One thousand replicates were used to calculate the bootstrap values. A phylogenetic tree was built using around 50 HCV sequences from other HCV isolates obtained from the HCV and GenBank databases.

### 2.9. Homology Modeling

The sequences of the NS5B polymerases of HCV GT4a were generated by the Sanger sequencing of 10 samples from responder and resistant patients, in addition to a reference (wild-type); these were used to construct 10 corresponding homology models. The sequence of the wild-type form of HCV GT4a polymerase was retrieved from the UniProtKB database (entry 4a (Y11604)) (www.uniprot.org, accessed on 8 September 2021). Eleven sequences (ten samples and a reference wild-type) were incorporated into the identification of templates with high identity, using the BLASTp server [24] against PDB databases. The search conditions for high-identity templates were optimized by setting an E-value cutoff of 0.0001. Based on the degree of similarity and the identity between the query sequences and templates, the most similar template from the search was selected to build models. The BLOSUM 62 weight matrix algorithm (alignment score), with a gap penalty and extension of 10 and 0.5, respectively, was implemented as the pairwise alignment algorithm. Target–template aligned sequences were uploaded to the HHpred server to generate target–template alignment profiles (http://toolkit.tuebingen.mpg.de/hhpred, accessed on 8 September 2021). Finally, the ten profiles, and the crystal structure of the selected template, were fed into the modeling software to generate the required 3D model structures. The final refinement of the models was completed using the Modrefiner server (https://zhanglab.ccmb.med.umich.edu/ModRefiner/, accessed on 9 September 2021).

### 2.10. 3D Structure Validation

The final refined 3D models of the HCV GT4a NS5B polymerase were validated using several servers, including Procheck to generate Ramachandran plots and the Verify protein 3D server to score the models [25].

### 2.11. Docking

The 10 optimized HCV NS5B polymerase models’ structures were used to perform the subsequent docking step. All the docking simulations were conducted using the MOE software [26]. The 10 enzymes and the sofosbuvir were prepared using the standard structure optimization protocol of the software. Moreover, the 10 enzymes were energy-minimized and the active sites were set as reported in the literature. The docking of sofosbuvir into the active sites of the 10 models was performed using triangular matcher as a placement method and London dg as a scoring function. The DS visualizer, available from Biovia Inc., was used to visualize and analyze the docking results through the generation of 2D interaction images.

### 2.12. Molecular Dynamics

#### 2.12.1. Molecular Dynamics for the Unbound Models

All the molecular dynamics in this work were assessed using the NAMD/MOE molecular interface. After solvating all 10 3D models using the TIP3P solvation model, the models were energy-minimized using the steepest descend algorithm under the AMBER12, extended Huckel theory (EHT) force field for a maximum of 50,000 steps [27]. The typical workflow of the NAMD simulation was applied for all of the MD simulations, starting by equilibrating every model for 5 ns, and all the well-equilibrated models were subjected to a production stage for 50 ns without any constraints using a time step of 1 fs. The generated trajectories were saved every 10 ps [28]. Furthermore, a constant temperature of 310 K and a constant pressure of 1 atm were maintained by Langevin dynamics and a Nose Hoover Langevin piston, respectively [29]. The particle mesh Ewald method with a grid point density of 0.92 Å was used to calculate the long-range electrostatics on a cubic box with periodic boundary conditions of 65 Å [30]. A cutoff of 12 Å for van der Waals interactions and a switching distance of 10 Å were used for production runs [30]. Finally, all the recorded trajectories were used to calculate the RMSD for all of the residues to judge the model’s stability.

#### 2.12.2. Molecular Dynamics for Sofosbuvir with Samples 10 (Responder) and 3 (Resistant)

The same conditions for simulating the unbound models were applied to simulate sofosbuvir in complex with samples 10 (responder) and 3 (resistant) for 50 ns. 

## 3. Results and Discussion

### 3.1. Sample Collection and Sequence Analysis

The presence of an HCV NS5B region was confirmed in 10 (29.4%) of the 34 samples using PCR. Five samples were from patients with virological failure, and the other five were from responder patients that were treated with a SOF-based regimen. After RNA extraction and cDNA synthesis, purification and sequencing were performed. For accurate genotyping, we sequenced the full-length NS5B, and the DNA Data Bank of Japan (DDBJ) was used for the submission of all of the ten positive HCV samples (Table 2). 

The Geno2Pheno webserver was implemented for detecting susceptible mutation sites. All the samples belonged to genotype 4a, with a similarity of nearly 90% to the reference sequence 4a (Y11604). This finding is consistent with other studies reporting from different governorates in Egypt, such as Damiette, Sharkia, Ismailia, and Alexandria [23,31,32,33,34,35]. HCV 4a is the most common subtype in south Egypt, followed by subtypes 1g, 4l, 4n, and 4o [32]. Subtype 4a is the most common subtype in the governorate of Ismailia, followed by subtypes 1g and 4o [34]. The most common subtype in the Sharkia governorate is 4a, followed by 4o, 1g, and 4n [31]. In the Alexandria governorate, the most common subtype is 4a, followed by 4m, 4o, 4n, and 4p [33]. Because of the small number of patients in the current study, subtypes other than 4a were not detected.

SOF, which targets the NS5B region, is extensively used in treating HCV patients in Egypt [36]. Due to the short period of SOF use, it is not yet known if treated patients may show post-treatment relapse [6]. Thus, we selected patients who failed to achieve the required sustained virologic responses (SVRs) after full therapeutic courses of treatment with SOF. The 3D structure of HCV NS5B polymerase has three domains: fingers, a palm, and a thumb. The active site of the palm domain comprises the residues 188–227 and 287–370 [37]. The important residues in the palm region contain D220, D225, G317, D318, and D319. The motif B residues G283, T286, T287, and N291, which participate in sugar selection by RNA-dependent RNA polymerase, are also highly conserved [37]. For the right palm, residues 360–370 are important for maintaining a proper secondary structure, forming motif E of the palm domain [37]. The presence of an amino acid substitution in these residues may hinder HCV polymerase activity [38]. First, one sample displayed the amino acid substitution D318Y, and two samples demonstrated a D319V mutation. Second, one sample had amino acid substitutions of G283S and T286H, and two samples showed amino acid substitutions of T287P/C and N291Y/T. Third, one sample showed the following amino acid substitutions: L360P, E361Q, L362E, I363T, T364R, S365G, C366T, S367P, S368V, N369E, and V370S. Two samples displayed L320G/R polymorphism (Table 3). This site confers low resistance to SOF treatment. Moreover, C316A/L polymorphism was present in two samples. Residues T532–R570 are responsible for regulating polymerase activity and lie in the NS5B C-terminus [38]. Sample 3 displayed eight amino acid substitutions, whereas sample 4 showed 21 amino acid substitutions: I5389L, A540P, G543A, R544K, T552N, G554A, Y555C, S556P, G557V, D559S, I560L, Y561T, H562S, S563K, V564P, S565L, H566G, A567E, R568I, P569R, and R570K. The presence of 21 amino acid substitutions indicates high variability in this region. A membrane anchor is formed from amino acids 571–591 in the NS5B C-terminus [38]. We recorded amino acid substitutions in three of the five samples, with 13 amino acid substitutions in this region: W571E, F572K, W573R, F574G, C575I, L576M, L577R, L578N, L579R, A580 Q/S, A581V, Y586F, and N590A. Most recent research has concentrated on a single catalytic site that can be made inactive by mutations affecting the S282 amino acid position, where S is replaced by T or, often, by C, which is mostly found in genotypes 1 and 2 [39]. Moreover, four of the five samples displayed substitutions T/C/R in position S282 (S282T/C/R) (Table 3). 

The therapeutic outcomes from targeting the HCV NS5B polymerase region are significantly affected by amino acid substitution. We constructed 3D structure models for all the samples, and these models were subjected to structural analysis to investigate the effects of amino acid substitution on the resistance to SOF.

### 3.2. Phylogenetic Analysis

Phylogenetic analysis was conducted on the NS5B sequences from the 10 analyzed samples. These isolates’ full-length NS5B sequences were associated with 50 HCV subtype 4 sequences, including the reference sequences and all the sequences from the Egyptian HCV isolates, to establish a neighbor-joining phylogenetic tree (Figure 1), with high bootstrap values. 

In terms of genotyping, gene sequencing analysis revealed that the 10 HCV isolates were all connected to HCV genotype 4, with variation in the subtypes. The phylogenetic tree showed two main branches: The first main branch included all our sequences, whereas the other sequences obtained from GenBank in addition to five sub-branches (all of them nearly genotype 4) showed differences when resistance analysis was conducted. The second main branch was divided into two sub-branches, including the genotype 4a (with a similarity to the closest reference above 90%), which included analysis of the full-length hepatitis C virus sequences obtained from GenBank. 

The sequences of the ten positive HCV samples were genotyped using the NCBI BLAST software as follows: 

The BLAST results for almost all of the samples, with accession numbers LC632938– LC632946, revealed that they were all HCV genotype 4a. Some of the most related sequences for those samples were MG454451, MG454460, MG60280, and MG453394, and all of these deposited samples were used in resistance analysis for patients with genotype 1–6 HCV infection treated with sofosbuvir/velpatasvir in phase III studies (ASTRAL-1, ASTRAL-2, and ASTRAL-3), with an identity of 92% [40]. KY608670: Hepacivirus C isolate 105, from France with genotype 4o NS5B gene, partial cds with an identity equal to 93% [41]. KY608675: Hepacivirus C isolate 1428 NS5B gene, partial cds, from USA with genotype 4o NS5B gene, with a 93% identity [42]. KY283130: Hepacivirus C strain EJ1, complete genome, with genotype 4a, isolate from Egypt with an 88% identity.

### 3.3. Homology Modeling

Homology modeling is critical because it is the only technique in the field of drug discovery that provides a 3D protein structure when a crystal or NMR structure is absent. Thus, the effect of sequence mutations in each sample on the resistance to SOF was assessed with 11 3D models constructed using the NS5B polymerase of genotype 1b PDB (3 hkw) as a template. This polymerase shows high identity and similarity with the target sequences (Figure 2 and Table 4). All of the servers used for validation indicated high-quality models. Ramachandran diagrams generated by Procheck [42] showed acceptable stereochemical properties for all the models (Figure 3). Furthermore, validation was ensured using Verify protein 3D, which was used to (1) assess the model’s accuracy by analyzing the compatibility of an atomic 3D model with its own amino acid sequence (1D), (2) calculate the percentage of residues with an average score > 0.2 (a residue score > 0.2 is considered reliable), and (3) conduct a final assessment of the 3D structure (pass, warning, or fail) [42]. The results of the verification were all “pass” for each of the generated 3D models.

### 3.4. Structural Analysis of the Generated Models

Polymerase enzymes replicate genomes into multiple copies. This activity requires certain structural features that enable the enzyme to perform its function. For instance, the enzyme must accommodate both the template and the replicate in its active site. Thus, certain dimensions in the active site are required. Such structural features and dimensions affect the enzyme activity and impact the binding of drugs to their targets. We reported distances between Glycine 188 and Threonine 227, and Threonine 287 and Valine 370, of 22.96 and 26.04, respectively, in the wild-type model (Figure 4 and Table 5). Variations in these distances may weaken the binding of SOF. For example, in sample 5, the distances between Gly188 and Thr227, and Thr287 and Val370, were 27.13 and 28.85, respectively (Figure 5). This increase in active site size probably prevents SOF from strongly interacting with the active site and consequently reduces its activity. Thus, we found that some amino acid substitutions in our samples potentially underlie the resistance to SOF via the alteration and disruption of the polymerase’s binding sites and structural features.

### 3.5. Docking

The current docking studies were conducted to provide insights into plausible mechanisms of sofosbuvir resistance in terms of binding interactions. In this context, sofosbuvir was docked in the active sites of the models. The results are highly consistent with the clinical outcomes, in which the binding of sofosbuvir with the resistant samples was tempered by the found mutations (Figure 6). This was evidenced by the lower docking scores achieved with sofosbuvir compared to the higher docking scores of sofosbuvir with the responder samples. Table 6 summarizes the docking results for sofosbuvir with the 10 samples.

### 3.6. Dynamic Simulation

#### 3.6.1. Molecular Dynamics for the Unbound Models

Dynamic simulation is of great value for drug discovery and biomedical studies, such as for finding new drugs for attractive targets [25], evaluating inhibitor strength, and explaining the effects of certain mutations on drug-resistance profiles. The latter use was targeted to achieve our primary goal of MD simulation. Accordingly, 10 molecular dynamic experiments of 50 ns were performed. Five resistant and five responder samples were assessed. Significant effects of the mutations on the response of HCV NS5B RNA-dependent RNA polymerase to DAAs were found. The MD simulations showed that the 3D models of the responder samples had greater stability than those of the resistant samples. The calculated RMSD ranged from 1.55 to 2.23 Å and from 3.20 to 3.87 Å for the responder and resistant samples, respectively (Figure 7 and Figure 8 and Appendix A and Table 7). The significant difference in the calculated RMSD for all of the residues of the most stable responder, sample 5 (1.8 Å), and the highest variable non-responder, sample 3 (3.4 Å), is depicted in Figure 9. The calculated RMSD of the responder sample was less than 2 Å throughout the 50 ns simulation, unlike the RMSD of the non-responder, which exceeded 2 Å. The reported mutations played a direct role in the increased dynamicity and flexibility of the HCV NS5B RNA-dependent RNA polymerase and diminished the ability of DAAs to form a stable complex with the enzyme binding site. In turn, this poor binding resulted in a failed virological response. Conversely, the 3D models from the responder samples showed relatively stable conformations, facilitating DAA–enzyme binding, yielding the desired efficacy.

#### 3.6.2. Molecular Dynamics for Sofosbuvir with Sample 10 (Responder) and Sample 3 (Resistant)

To reinforce the docking results, the predicted binding poses of sofosbuvir with samples 10 (responder) and 3 (resistant) were simulated for 50 ns. The results were perfectly aligned with the clinical and docking results, as indicated by the RMSD calculations (Figure 10). The RMSD for sofosbuvir in complex with sample 10 was less than 2 Å throughout the simulation, indicating stable binding and highlighting the ability of sofosbuvir to inhibit the responder samples. By contrast, the RMSD of sofosbuvir in complex with sample 3 exceeded 3 Å, highlighting unstable binding, which explains the resistance to therapy with sofosbuvir. 

## 4. Conclusions

We analyzed Egyptian HCV patients with virological failure by sequencing the NS5B region and recording amino acid substitutions. According to our findings, the most common HCV genotype among Egyptian patients was 4a. In addition, the findings demonstrate the relationship between the amino acid substitution and resistance to a SOF-based regimen in the NS5B of HCV genotype 4a-infected patients. To the best of our knowledge, this is the first study to report the application of molecular dynamic simulations to assess the resistance to sofosbuvir attributable to mutations in NS5B polymerase in genotype 4a. Collectively, the findings of this study encourage designing a specific treatment protocol for HCV genotype 4a-infected patients, an infection that is dominant in Egypt, to avoid resistance. Finally, the limitation of this study was the minimal sample size due to funding constraints. More research with larger sample size is required to verify the current findings. 

## Figures and Tables

**Figure 1 microorganisms-10-00679-f001:**
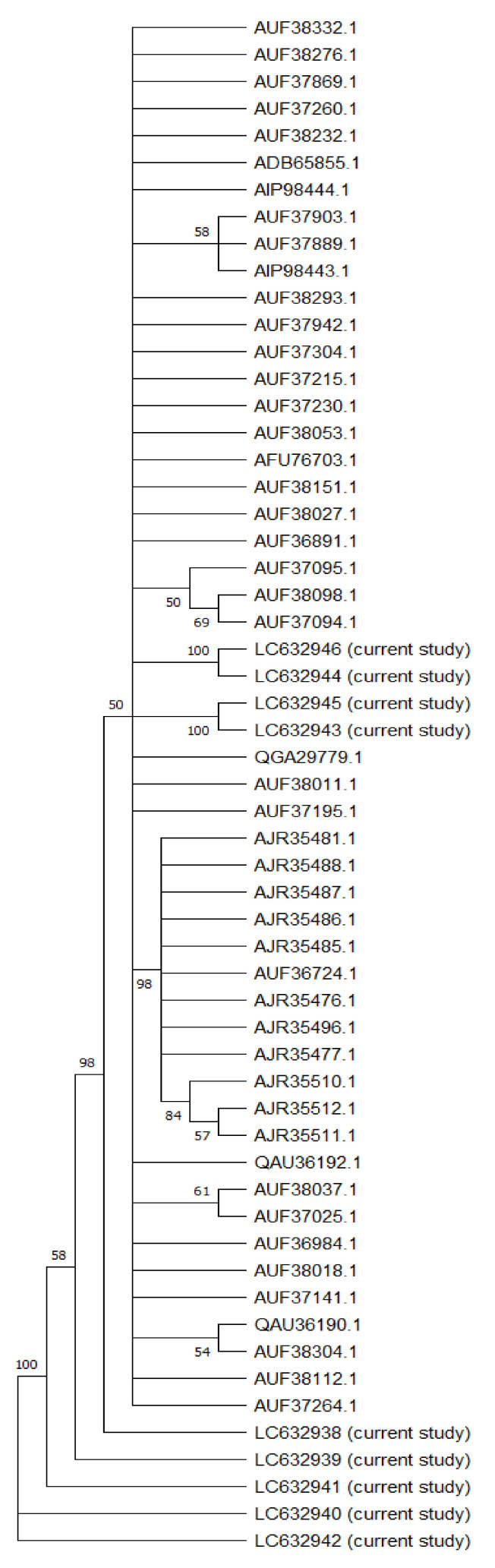
Neighbor-joining phylogenetic tree of the full-length sequence of NS5B of several HCV genotype 4 representatives. Bootstrap values based on 1000 replicates are shown next to the branches; bootstrap values of more than 50% only are shown. Sequences are labeled to the right of each branch in the order of GenBank accession number and isolate name.

**Figure 2 microorganisms-10-00679-f002:**
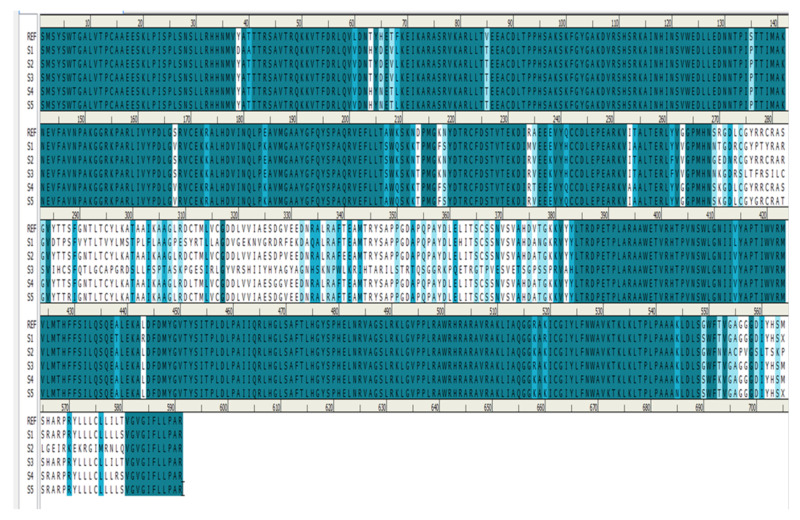
Sequence alignment between the reference (R) (Y11604) and the five resistant samples (S1–S5).

**Figure 3 microorganisms-10-00679-f003:**
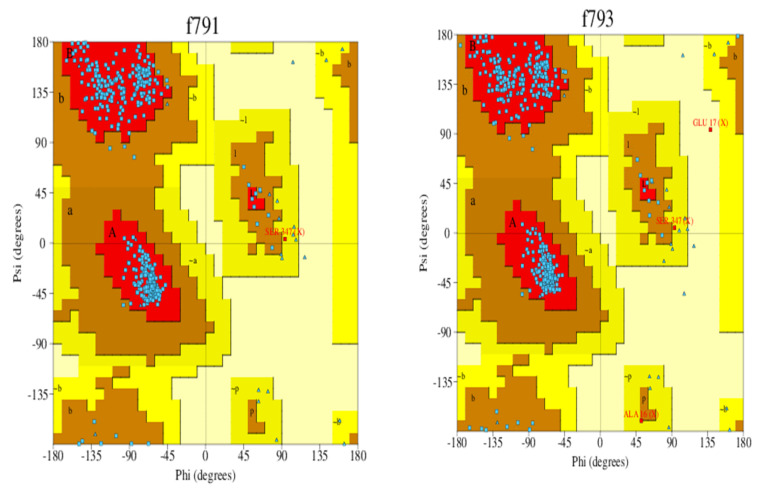
Ramachandran plot of the 3D models of HCV NS5B genotype 4a-resistant samples and reference exceeding 91.4% of the residues in the favored regions and around 5% of residues in the allowed regions. F791 (reference), f793 (sample 1), f803 (sample 2), f804 (sample 3), f805 (sample 4), and f806 (sample 5).

**Figure 4 microorganisms-10-00679-f004:**
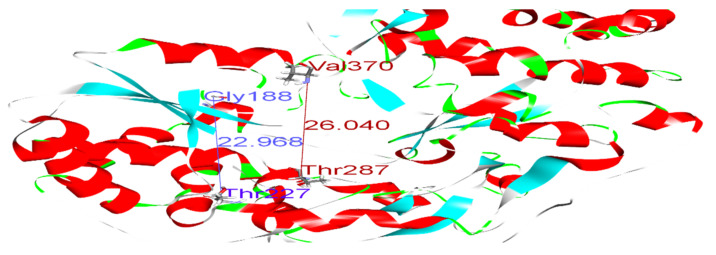
Distances between Gly188 and Thr227 (blue), and between Thr287 and Val370 (dark red), in the wild-type model.

**Figure 5 microorganisms-10-00679-f005:**
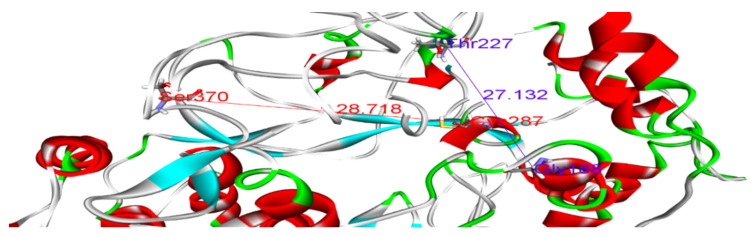
Distances between Gly188 and Thr227 (blue), and between Cys287 and Ser370 (dark red), in the sample 5 model.

**Figure 6 microorganisms-10-00679-f006:**
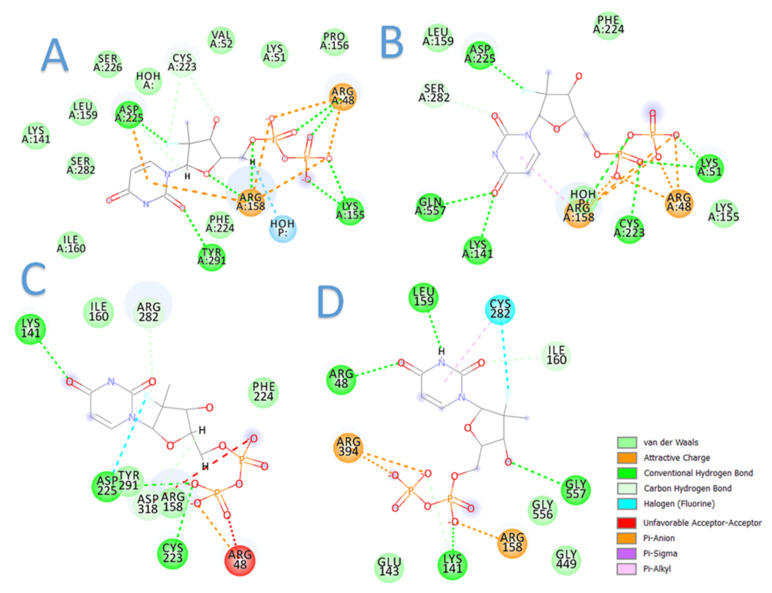
Two-dimensional diagrams of sofosbuvir interaction with HCV GT4a NS5B: (**A**) sample 9 (responder), (**B**) sample 10 (responder), (**C**) sample 3 (resistant), and (**D**) sample 5 (resistant).

**Figure 7 microorganisms-10-00679-f007:**
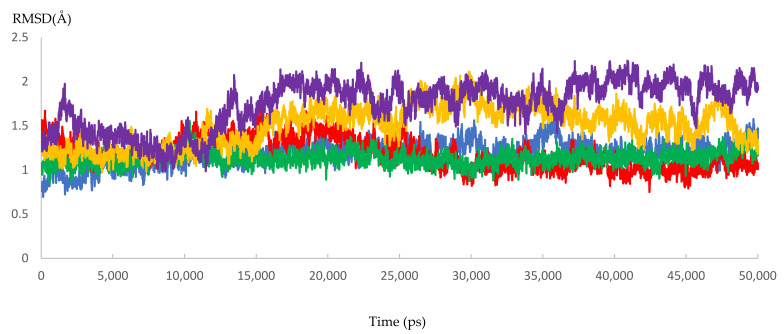
The RMSD of five dynamic simulation experiments for the HCV 4a NS5B responder samples. The green color represents sample 6, red represents sample 7, blue represents sample 8, yellow represents sample 9, and purple represents sample 10.

**Figure 8 microorganisms-10-00679-f008:**
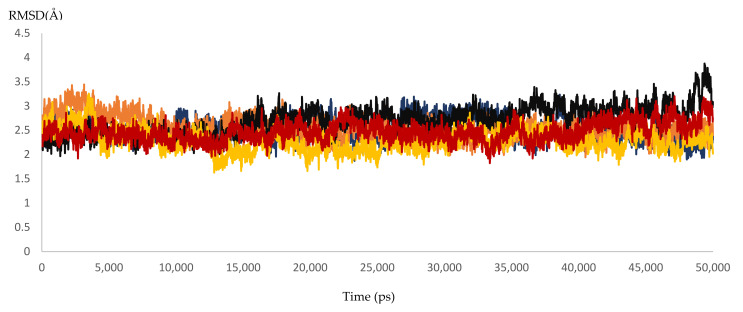
The RMSD of five dynamic simulation experiments for HCV 4a NS5B resistant samples. The blue color represents sample 1, brown represents sample 2, black represents sample 3, yellow represents sample 4, and red represents sample 5.

**Figure 9 microorganisms-10-00679-f009:**
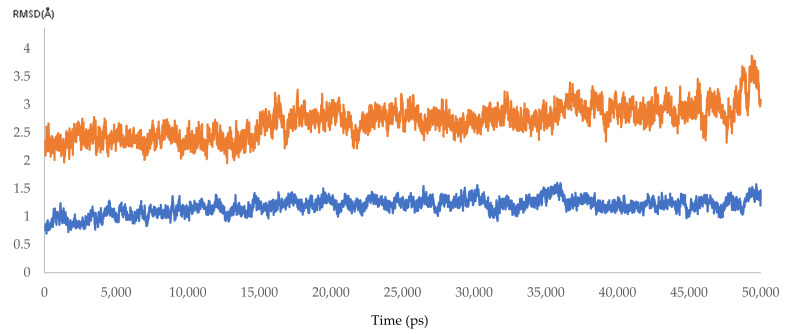
The RMSD of HCV 4a NS5B, comparing the highest resistant and responder samples (3.4 and 1.8 Å).

**Figure 10 microorganisms-10-00679-f010:**
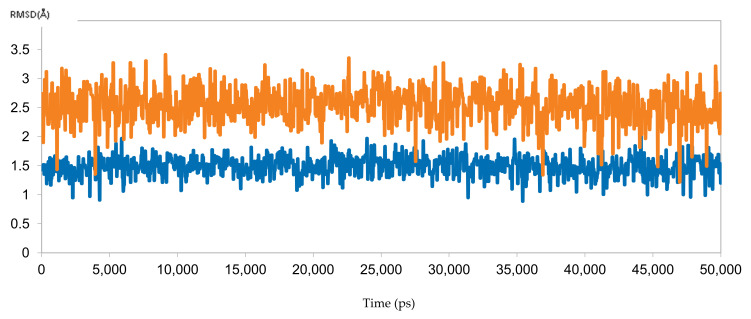
The RMSD of two dynamic simulations of sofosbuvir in complex with HCV 4a NS5B, comparing responder sample 10 (blue) and resistant sample 3 (brown).

**Table 1 microorganisms-10-00679-t001:** The primers used in this work.

Primer Name	Sequence	Reference
G4R1F	TWGTAACACCCTGTGCDGCTGAAG	This study
G4R1R	CCGCAAACCARCATAGTGCAYT	This study
G4R2F	CTGAGAGACTGCACYATGCTYGT	This study
G4R2R	CCCTARGGTCGGAGTGTTAAGCT	This study

F, forward primers; R, reverse primers.

**Table 2 microorganisms-10-00679-t002:** Accession numbers for ten novel genotypes.

Accession Number	Sample No.
LC632938	Sample 1
LC632939	Sample 2
LC632940	Sample 3
LC632941	Sample 4
LC632942	Sample 5
LC632943	Sample 6
LC699305	Sample 7
LC632944	Sample 8
LC632945	Sample 9
LC632946	Sample 10

**Table 3 microorganisms-10-00679-t003:** Amino acid variance in the NS5B region of nine HCV isolates.

4a ED43 Y11604		LC632938	LC632939	LC632940	LC632941	LC632942	LC632943	LC632944	LC632945	LC632946
**235**	A	V	V	V	V	V	V	T	V	T
241	Q	Q	H	H	H	H	H	R	R	R
254	T	T	T	T	T	T	A	A	A	A
261	Y	Y	F	Y	F	F	Y	Y	Y	Y
267	H	H	L	H	H	H	H	H	H	H
270	R	R	L	T	E	K	K	K	K	K
275	G	G	G	G	G	L	G	G	G	G
**282**	**S**	**T**	**R**	**R**	**R**	**C**	**S**	**S**	**S**	**S**
285	Y	Y	Y	D	Y	I	Y	Y	Y	Y
293	L	L	L	L	L	G	L	L	L	L
303	I	I	I	F	I	F	I	I	I	I
307	G	G	G	G	G	A	G	R	A	R
309	R	R	R	E	R	K	R	R	R	R
322	V	V	V	K	V	H	V	V	V	V
327	D	D	G	D	D	Y	D	D	D	D
333	N	N	N	A	N	N	N	N	N	N
336	L	L	L	L	L	K	L	L	L	L
**360**	**L**	**L**	**L**	**L**	**L**	**P**	**L**	**L**	**L**	**L**
**361**	**E**	**E**	**E**	**E**	**E**	**Q**	**E**	**E**	**E**	**E**
**363**	**I**	**I**	**I**	**I**	**I**	**T**	**I**	**I**	**I**	**I**
**365**	**S**	**S**	**S**	**S**	**S**	**G**	**S**	**S**	**S**	**S**
**367**	**H**	**S**	**A**	**S**	**S**	**P**	**S**	**S**	**S**	**S**
**370**	**V**	**V**	**V**	**V**	**V**	**S**	**V**	**V**	**V**	**V**
**552**	**T**	**T**	**K**	**T**	**T**	**T**	**T**	**M**	**T**	**M**
**555**	**A**	**A**	**A**	**A**	**A**	**A**	**A**	**S**	**A**	**S**
**557**	**G**	**G**	**G**	**G**	**G**	**G**	**G**	**Q**	**G**	**Q**
**559**	**D**	**D**	**D**	**D**	**D**	**D**	**D**	**H**	**D**	**H**
**561**	**Y**	**Y**	**Y**	**Y**	**Y**	**Y**	**Y**	**S**	**Y**	**S**
**566**	**H**			**Q**			**A**	**Q**	**A**	**Q**
**569**	**P**			**T**			**R**	**T**	**R**	**T**
**570**	**R**			**P**		**P**	**Y**	**P**	**Y**	**P**
**573**	**L**			**R**		**L**	**L**	**R**	**L**	**L**
**574**	**L**			**P**		**Q**	**L**	**G**	**L**	**C**
**577**	**L**			**Q**				**R**		**T**
**579**	**P**			**C**					**R**	**Y**
**580**	**T**			**S**			**S**			**P**
**590**	**A**			**L**			**A**			

**Table 4 microorganisms-10-00679-t004:** Sequence identity and similarity between the template and five resistant samples.

Name	Identity	Similarity
Reference 4a (Y11604)	78%	1.28
Sample 1	78%	1.28
Sample 2	76%	1.26
Sample 3	72%	1.17
Sample 4	75%	1.25
Sample 5	67%	1.07

**Table 5 microorganisms-10-00679-t005:** Structural analysis and distances measured in each generated model.

Name	Distance from Gly188 to Thr227	Distance from Thr287 to Val370
Reference 4a (Y11604)	22.96 Å	26.04 Å
Sample 1	25.22	27.01
Sample 2	23.99	25.76
Sample 3	24.57	24.70
Sample 4	24.55	26.19
Sample 5	27.13	28.85

**Table 6 microorganisms-10-00679-t006:** The docking scores of sofosbuvir with the 10 models of HCV GT4a NS5B.

Sample No.	Docking Score (KCal/Mole)
Sample 1 (resistant)	–16.5
Sample 2 (resistant)	–17.3
Sample 3 (resistant)	–15.8
Sample 4 (resistant)	–17.4
Sample 5 (resistant)	–16.2
Sample 6 (responder)	–22.7
Sample 7 (responder)	–21.9
Sample 8 (responder)	–22.3
Sample 9 (responder)	–22.9
Sample 10 (responder)	–23.6

**Table 7 microorganisms-10-00679-t007:** Average RMSD for 10 molecular dynamic samples for 50 ns (five resistant and five responder samples).

Sample Number	Average (Å) ± SD	Status
S1	2.524207 ± 0.1	Resistant
S2	2.55801 ± 0.11	Resistant
S3	2.700728 ± 0.08	Resistant
S4	2.293318 ± 0.09	Resistant
S5	2.471093 ± 0.12	Resistant
S6	1.189008 ± 0.06	Responder
S7	1.190665 ± 0.07	Responder
S8	1.132385 ± 0.07	Responder
S9	1.505865 ± 0.05	Responder
S10	1.745415 ± 0.08	Responder

## Data Availability

Accession numbers for ten novel genotypes: LC632938, LC632939, LC632940, LC632941, LC632942, LC632943, LC699305, LC632944, LC632945, LC632946.

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
