# Peer review of "Identifying Different Mutation Sites Leading to Resistance to the Direct-Acting Antiviral (DAA) Sofosbuvir in Hepatitis C Virus Patients from Egypt"

_microorganisms, 2022, doi:10.3390/microorganisms10040679_

Round 1

Reviewer 1 Report

The authors have addressed the reviewer's comments. 

Author Response

Thanks a lot for your time.

Reviewer 2 Report

I would like to thank the authors for addressing my comments on the manuscript and many of my concerns have been addressed.  The molecular docking work strengthens the conclusions.  Several of my concerns have not been fully addressed however and must be prior to me recommending this paper for publication.  

 - The authors explained that sequence 7 has not been deposited, however they have not explained why.  This (and the explanation) should be included in the manuscript for transparency.

- I realise that RMSD plots are often represented as a trace, however my suggestion that these be represented as histograms was so that the significance of the changes could be established, as well as measuring the equilibration of the simulations.  For example, a histogram taken over the first 10ns of the resistant sample from figure 9 will have a very different average than one taken over the last 10ns.  This simulation has clearly not equilibrated.  While I am happy for this plot to be included given the difference between the samples, this should definitely be commented on in the manuscript.  The averages are presented in table 7, but it would be a good to include the standard deviation in this table as well.  

- Many of the figures are still of low quality, and should be improved prior to publication.  The units of the x axis of figures 7-10 should be labelled as ps (lower case).  Figure 10 has a shadow, while others have a border in places.  The top of figure 3 is cut off, the aspect ratio of figure 2 is off, and the phylogenetic tree has too low resolution to read correctly.  The caption also mentions that bootstrap values of less than 50% are not shown, however many values shown are less than 50%.  Perhaps exporting the figures as pdfs would help? Figure 4 is still not easy to read.

- I assume that the gel electrophoresis plots are still in the SI (I can't access the file).  If they are relevant enough to be included with the manuscript, they should be mentioned (even if it is only a description after the acknowledgements).  I don't think they should be excluded if they are part of the study, as this is important for transparency.               

Author Response

Comment: The authors explained that sequence 7 has not been deposited, however, they have not explained why.  This (and the explanation) should be included in the manuscript for transparency.

Response:  I appreciate your perceptive comments. I submitted the sequence "7" to DDBJ earlier, but it was returned due to a computational problem. I have been resubmitted and the accession number is LC699305. I will add in the galley proof version.

Comment: I realise that RMSD plots are often represented as a trace, however my suggestion that these be represented as histograms was so that the significance of the changes could be established, as well as measuring the equilibration of the simulations.  For example, a histogram taken over the first 10ns of the resistant sample from figure 9 will have a very different average than one taken over the last 10ns.

Response: We thank the reviewer for such a suggestion; indeed histogram is more representative than trace illustration. However, it is not applicable to plot millions of conformations generated from the molecular dynamics as histogram, the only way unfortunately is the trace plot. Yet we plotted the average RMSD values as histogram which will be provided as supporting information.

Also, as the reviewer mentioned the equilibration, all the systems were equilibrated prior to commencing the dynamic production, in addition, the RMSD plotting was smooth without any broken structures.    

Comment: This simulation has clearly not equilibrated.  While I am happy for this plot to be included given the difference between the samples, this should definitely be commented on in the manuscript.  

Response: Thanks for your precise note, indeed all the simulations were equilibrated for 5ns prior to the molecular dynamics production.

Comment: The averages are presented in table 7, but it would be a good to include the standard deviation in this table as well.

Response: We agree with that, and it was done as suggested by the reviewer.

Comment: Many of the figures are still of low quality and should be improved prior to publication.

 Response: Thank you for pointing this out. The reviewer is correct, and we addressed it in the revised manuscript, as all images will be provided as separate files with good quality.

Comment: The units of the x axis of figures 7-10 should be labelled as ps (lower case).  

Response: Thanks for pointing this out. Done as suggested by the reviewer.

Comment: Figure 10 has a shadow, while others have a border in places.

Response: Thanks for your precise note and all figures were amended to have a uniform representation.

Comment: The top of figure 3 is cut off, the aspect ratio of figure 2 is off, and the phylogenetic tree has too low resolution to read correctly.

Response: Thank you so much for your minute observation and valuable comment. Done as suggested by the reviewer

Comment: The caption also mentions that bootstrap values of less than 50% are not shown, however many values shown are less than 50%.  Perhaps exporting the figures as pdfs would help?

Response: Thank you for pointing this out. The reviewer is correct, and we have fixed it in the revised manuscript.  

Comment: Figure 4 is still not easy to read.

Response: Thank you so much for your minute observation, and I amended it as suggested by the reviewer.

Comment: I assume that the gel electrophoresis plots are still in the SI (I can't access the file).  If they are relevant enough to be included with the manuscript, they should be mentioned (even if it is only a description after the acknowledgements).  I don't think they should be excluded if they are part of the study, as this is important for transparency.    

Response:  As suggested by the reviewer, we have addressed it in the revised manuscript, and deposited it in the SI.

Round 2

Reviewer 2 Report

I am satisfied that the authors have now sufficiently addressed all of my concerns and can recommend this paper for publication.

This manuscript is a resubmission of an earlier submission. The following is a list of the peer review reports and author responses from that submission.

Round 1

Reviewer 1 Report

The authors presented a work identifying mutations associated with resistance to SOF in genotype 4 HCV isolated from patients. The are several issues that must be corrected to improve the overall scientific soundness and quality of ms.

This paper could be separated into two major outcomes: 1) molecular virology (Genotype determination and mutant resistance to DAA) and 2) computational chemistry (mainly focused on drug resistance). I will comment on each one in the following... 

There is a major flaw in this paper associated with the use of computational tools to determine drug resistance. Computational tools are interesting and could help to understand/ be used as a first approach when are correctly applied under a strong hypothesis. However, this is not the case. The authors said that their MD results support their findings. The major finding is related to protein stability (in terms of RMSD, MD simulations results). However, Drug efficacy cannot be directly associated with this parameter. You are evaluating the OVERALL stability of the protein, not the stability of the biding site.

So, based on your results... Could you said that the protein is less "stable", thus, it is less functional? The answer is: No, the virus still replicating...

Do the changes in stability affect the binding of the drug? If yes, how? The only way to answer that question is to evaluate the interaction of the ligand with each of your mutants via docking and further mm/PB(Gb)sa analysis. If not, the conclusion is a flaw.  This is what the reader is expecting when see a title of drug resitance mutations, drugs, and pharmacological target.

1) The title is not clear and does not reflect the ms content. Thus, the author must modify the title.

2) The abstract is not clear. The ideas appear without a clear flow. Also, the author said in the abstract: to support our hypothesis... without defining a hypothesis. 

3) The introduction needs to be re-write. It is confusing, the author talks about epidemiology, then about viral biology, and then goes back to epidemiology... Make it clear for the reader!

4) The methodology needs to be ordered. 2.3 and 2.5 should be the following points. 
Did you use random, forward, and reverse primers, altogether in the same cDNA reaction?

if the virus is a positive sense (5' to 3') Why do you use forward primer for cDNA (5' to 3')?

5) Results
All 20 samples have a viral load. Why do you be not able to amplify all?
The authors said: conserved residues for sugar selection by NS5B... What do you mean by this? explain
When the author enumerates all the mutants the paper gets messy. Use the table as a resource and highlight the most important mutations in the text. 
Phylogenetic analysis... Where is the discussion for these results? the authors said nothing about it. Is it needed? Highlight your samples in the tree.
The author focused their structural analysis on distances. The authors must prove that 1) these distances produce less efficient polymerase (if you want to talk about protein stability) and 2) these distances are traduced in a weakening interaction to SOF (if you want to talk about conformational changes associated with lower response to SOF).

6) The conclusions of the work are not supported by the results.

Reviewer 2 Report

In this work the authors combine sequencing data taken from patients, with molecular dynamics modelling to try to predict/understand the efficacy of treatment using sofosbuvir.  This is an important an interesting topic, which could be of interest to the reader.  However, there are many shortcomings with the manuscript. 

1) The methods lack a lot of detail & specificity about how the research was performed.  For example, the patients are described as having viral loads from 307,000 to 18,969,000 IU/L with no description on how this was established or references to other works where this has been measured.  It is frequently mentioned that ten samples were used, while only nine sequences appear to have been deposited with NCBI.  This discrepancy is not commented on.  The samples on the NCBI database also vary greatly in length, hence calculation of the protein sequences would need to account for this, which does not appear to have occurred, or has not been mentioned.  It appears sequence 7 has not been deposited.  It is also not clear which sequences are discussed later in the manuscript.  Does sample=seq? The phylogenetic tree is presented as a cladogram, which is slightly misleading and inappropriate for genomic data.  On what basis were the 50 sequences selected?  

2) The modelling work also lacks rigour.  While the WT protein is one form of control, would substitution of mutations not associated with sofosbuvir resistance also produce changes in amino-acid distance/angles?  Perhaps mutations not discovered in this study could be tested.  Given that RMSD results are presented, could the distances presented earlier in the manuscript not be measured in those simulations to give an average distance?  The RMSD results could easily be presented as histograms, rather than long traces.  It does not appear that all simulations have equilibrated e.g. sample 55 & 54 in figure 6, or both traces in figure 8, which are still drifting.    

3) The manuscript is not well presented & poorly organised.  I appreciate that if the authors first language is not English that some rewriting may be required.  However the figures are often unreadable (Figs 2, 4 & 5 and the corresponding SI figures), or poorly presented (Fig 6-8).   The labelling of the data in figures 6 & 7 is also inconsistent with the rest of the manuscript.  The gel electrophoresis plot presented in the supporting files is not mentioned in the manuscript.  Tables 1 & 2 belong in supplementary information, not in the main text.  Table 2 is not mentioned, and the reference to table 1 appears to actually refer to table 3.               

Overall the manuscript is not ready for publication.  It may have been more scientifically sound to split the work into two publications (which would both require significantly more work), where one work focused on the phylogeny & biological context of the new sofosbuvir-resistant genotypes, and another work which looked at simulating the structural basis of sofosbuvir resistance in NS5B.  By trying to do both the manuscript has performed neither task satisfactorily.